# Scaling Gaussian Processes for Learning Curve Prediction via Latent Kronecker Structure

**Jihao Andreas Lin**[*,1,2,3]     **Sebastian Ament**[1]     **Maximilian Balandat**[1]     **Eytan Bakshy**[1]

[1]Meta     [2]University of Cambridge     [3]Max Planck Institute for Intelligent Systems

## Abstract

A key task in AutoML is to model learning curves of machine learning models jointly as a function of model hyper-parameters and training progression. While Gaussian processes (GPs) are suitable for this task, naïve GPs require $\mathcal{O}(n^3m^3)$ time and $\mathcal{O}(n^2m^2)$ space for $n$ hyper-parameter configurations and $\mathcal{O}(m)$ learning curve observations per hyper-parameter. Efficient inference via Kronecker structure is typically incompatible with early-stopping due to missing learning curve values. We impose *latent Kronecker structure* to leverage efficient product kernels while handling missing values. In particular, we interpret the joint covariance matrix of observed values as the projection of a latent Kronecker product. Combined with iterative linear solvers and structured matrix-vector multiplication, our method only requires $\mathcal{O}(n^3 + m^3)$ time and $\mathcal{O}(n^2 + m^2)$ space. We show that our GP model can match the performance of a Transformer on a learning curve prediction task.

## 1 Introduction

Most modern machine learning (ML) models are trained with iterative methods, giving rise to learning curves which enable an analysis of the model quality as the training progresses. Being able to predict learning curves accurately based on results from partial training facilitates decisions about whether to continue training or to stop early, such that compute resources can be used more efficiently. This can substantially accelerate model exploration and improve the efficiency of AutoML algorithms.

Existing work on learning curve prediction considered Gaussian processes (GPs) [Swersky et al., 2014, Klein et al., 2020, Wistuba et al., 2022], parametric functions [Domhan et al., 2015, Klein et al., 2017, Kadra et al., 2023], and Transformers [Adriaensen et al., 2023, Rakotoarison et al., 2024] (see Appendix A for details). In this paper, we revisit the idea of a joint GP over hyper-parameters and learning curves. In principle, it is possible to define a GP on the product space of hyper-parameters and learning curves, but, for $n$ hyper-parameter configurations and $\mathcal{O}(m)$ learning curve observations per hyper-parameter, this naïve approach requires $\mathcal{O}(n^3m^3)$ time and $\mathcal{O}(n^2m^2)$ space, which quickly becomes infeasible. An efficient approach is to impose Kronecker structure in the joint covariance matrix [Bonilla et al., 2007, Stegle et al., 2011, Zhe et al., 2019]. However, this requires complete learning curves for each hyper-parameter configuration, which is incompatible with early-stopping.

Herein, we leverage *latent Kronecker structure*, which enables efficient inference despite missing entries by obtaining the joint covariance matrix of observed values from the full Kronecker product via lazily evaluated projections. With suitable posterior inference techniques, this reduces the asymptotic time complexity from $\mathcal{O}(n^3m^3)$ to $\mathcal{O}(n^3+m^3)$, and space complexity from $\mathcal{O}(n^2m^2)$ to $\mathcal{O}(n^2+m^2)$ compared to the naïve approach. We demonstrate this superior scalability empirically, and show that our GP with generic stationary kernels and 10 free parameters can match the performance of a specialized Transformer model with 14.69 million parameters on a learning curve prediction task.

---

[*]Work done during an internship at Meta. Correspondence to: `jal232@cam.ac.uk`

Workshop on Bayesian Decision-making and Uncertainty, 38th Conference on Neural Information Processing Systems (NeurIPS 2024).

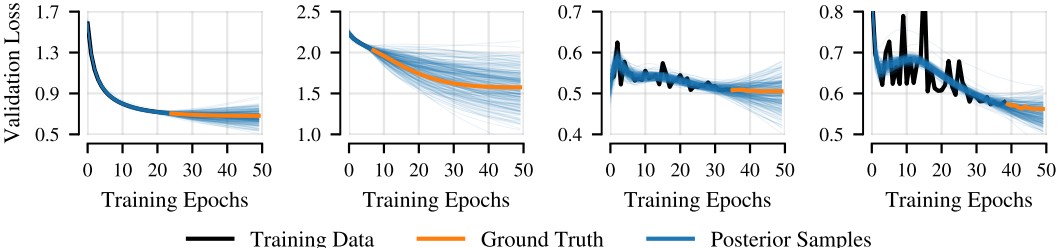

Figure 1: Learning curve predictions on the Fashion-MNIST data from the LCBench benchmark [Zimmer et al., 2021]. The GP is fit to 16 partially observed learning curves (black). Their ground truth continuations (orange) are contained within the spread of posterior samples (blue). A typical learning curve which is observed close to convergence is predicted with confidence (left). Observing a smaller fraction of the learning curve leads to increased uncertainty in the prediction (left middle). The model also adapts well to less common noisy and spiky learning curves (right).

## 2 Gaussian Processes with Latent Kronecker Structure

We consider the problem of defining a GP on the product space of hyper-parameter configurations $\mathbf{x} \in \mathbb{R}^d$ and learning curve progression $t \in \mathbb{R}$, namely $f : \mathbb{R}^d \times \mathbb{R} \to \mathbb{R}$, where $f \sim \mathrm{GP}(\mu, k)$ with mean function $\mu$ and kernel $k$. Throughout this paper, we suppress the dependence on kernel hyper-parameters for brevity of notation, set $\mu = 0$ without loss of generality, and assume homoskedastic observation noise $\sigma^2$. For an introduction to GPs, we refer to Rasmussen and Williams [2006].

The simplest way to address this problem is to define a kernel directly on the product space, which results in a joint covariance $\mathrm{Cov}(f(\mathbf{x}, t), f(\mathbf{x}', t')) = k((\mathbf{x}, t), (\mathbf{x}', t'))$. However, this quickly runs into scalability issues. Assuming we evaluate $n$ hyper-parameter configurations $\mathbf{X} := \{\mathbf{x}_i\}_{i=1}^n$ at $m$ progressions $\mathbf{t} = \{t_1, \dots, t_m\}$, observing learning curves $\mathbf{Y} := \{\mathbf{y}_i \in \mathbb{R}^m\}_{i=1}^n$, the joint covariance matrix requires $\mathcal{O}(n^2 m^2)$ space, and computing its Cholesky factorization takes $\mathcal{O}(n^3 m^3)$ time.

**Latent Kronecker Structure**  A common way to improve the scalability of GPs on product spaces is to introduce Kronecker structure [Bonilla et al., 2007, Stegle et al., 2011, Zhe et al., 2019]. In particular, one may define a product kernel $k((\mathbf{x}, t), (\mathbf{x}', t')) = k_1(\mathbf{x}, \mathbf{x}') \, k_2(t, t')$, where $k_1$ only acts on hyper-parameter configurations $\mathbf{x}$ and $k_2$ only considers learning curve progressions $t$. When applied to the observed data, the resulting joint covariance matrix $\mathbf{K}$ has Kronecker structure,

$$\mathop{\mathbf{K}}_{nm \times nm} = k((\mathbf{X} \times \mathbf{t}), (\mathbf{X} \times \mathbf{t})) = k_1(\mathbf{X}, \mathbf{X}) \otimes k_2(\mathbf{t}, \mathbf{t}) = \mathop{\mathbf{K}_1}_{n \times n} \otimes \mathop{\mathbf{K}_2}_{m \times m} \, ,$$

which can be exploited by expressing the eigenvalue decomposition of $\mathbf{K}$ in terms of $\mathbf{K}_1$ and $\mathbf{K}_2$. This reduces the asymptotic time complexity to $\mathcal{O}(n^3 + m^3)$ and space complexity to $\mathcal{O}(n^2 + m^2)$.

Unfortunately, the joint covariance matrix only has Kronecker structure if each $\mathbf{x}_i$ is evaluated at every $t_j$, that is, each learning curve must be fully observed, which conflicts with early-stopping. However, *the joint covariance matrix over partially observed learning curves is a submatrix of the latent Kronecker product*, and the former can be selected from the latter using a projection matrix $\mathbf{P}$,

$$\mathbf{K}_{\mathrm{joint}} = \mathbf{P} \, \mathbf{K}_{\mathrm{latent}} \, \mathbf{P}^\mathsf{T} = \mathbf{P}(\mathbf{K}_1 \otimes \mathbf{K}_2)\mathbf{P}^\mathsf{T},$$

where $\mathbf{P}$ is constructed from the identity matrix by removing rows corresponding to missing values. In practice, projections can be implemented efficiently via slice indexing without instantiating $\mathbf{P}$.

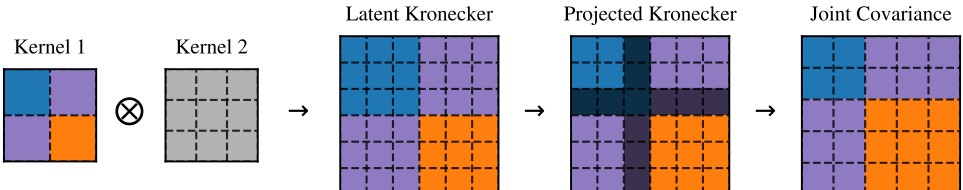

Figure 2: Selecting the joint covariance matrix using projections of the latent Kronecker product after observing $\{(\mathbf{x}_1, t_1), (\mathbf{x}_1, t_2), (\mathbf{x}_2, t_1), (\mathbf{x}_2, t_2), (\mathbf{x}_2, t_3)\}$, two learning curve values from a first hyper-parameter configuration (blue) and three values from a second configuration (orange).

**Efficient Inference with Iterative Methods**   As a result of introducing projections, the eigenvalues and eigenvectors of $\mathbf{K}_{\text{joint}}$ cannot be expressed in terms of eigenvalues and eigenvectors of $\mathbf{K}_1$ and $\mathbf{K}_2$ anymore, which prevents the use of Kronecker structure for efficient matrix factorization. However, despite projections, the Kronecker structure can still be leveraged for fast matrix multiplication,

$$(\mathbf{A} \otimes \mathbf{B}) \operatorname{vec}(\mathbf{C}) = \operatorname{vec}(\mathbf{B}\mathbf{C}\mathbf{A}^{\mathsf{T}}) \;\rightarrow\; \mathbf{P}(\mathbf{A} \otimes \mathbf{B})\mathbf{P}^{\mathsf{T}} \operatorname{vec}(\mathbf{C}) = \mathbf{P} \operatorname{vec}(\mathbf{B} \operatorname{vec}^{-1}(\mathbf{P}^{\mathsf{T}} \operatorname{vec}(\mathbf{C}))\mathbf{A}^{\mathsf{T}}),$$

where, in practice, $\operatorname{vec}$ and $\operatorname{vec}^{-1}$ correspond to reshaping, $\mathbf{P}^{\mathsf{T}} \operatorname{vec}(\mathbf{C})$ amounts to zero padding, and $\mathbf{P}$ is slice indexing. This facilitates efficient GP inference via *iterative methods*, which only rely on matrix-vector multiplication (MVM) to compute solutions to systems of linear equations [Gardner et al., 2018, Wang et al., 2019]. Leveraging latent Kronecker structure and lazy kernel evaluations, iterative methods require $\mathcal{O}(n^2m + nm^2)$ time and $\mathcal{O}(n + m)$ space. This is similar to structured kernel interpolation (SKI) [Wilson and Nickisch, 2015], an inducing point approximation that creates Kronecker structure by placing inducing points on regular grids. Compared to SKI, our approach only applies a product structure to separate hyper-parameters $\mathbf{x}$ from progressions $t$, and permits *exact* MVMs as long as progressions are logged on a shared grid, such as epochs.

**Posterior Samples via Matheron's Rule**   Maddox et al. [2021] proposed to draw posterior samples efficiently by exploiting Kronecker structure via Matheron's rule [Wilson et al., 2020, 2021], which expresses a posterior sample in terms of a transformed prior sample,

$$(f|\mathbf{Y})(\cdot_{\mathbf{x}}, \cdot_t) = f(\cdot_{\mathbf{x}}, \cdot_t) + (k_1(\cdot_{\mathbf{x}}, \mathbf{X}) \otimes k_2(\cdot_t, \mathbf{t}))(\mathbf{K}_1 \otimes \mathbf{K}_2 + \sigma^2\mathbf{I})^{-1}(\operatorname{vec}(\mathbf{Y}) - f((\mathbf{X} \times \mathbf{t})) - \boldsymbol{\epsilon}),$$

where $(f|\mathbf{Y})$ is the posterior sample, $f$ is the prior sample, $f((\mathbf{X} \times \mathbf{t}))$ is its evaluation at the training data, and $\boldsymbol{\epsilon} \sim \mathcal{N}(\mathbf{0}, \sigma^2\mathbf{I})$. To support *latent* Kronecker structure, we introduce projections,

$$f(\cdot_{\mathbf{x}}, \cdot_t) + \underbrace{(k_1(\cdot_{\mathbf{x}}, \mathbf{X}) \otimes k_2(\cdot_t, \mathbf{t}))\mathbf{P}^{\mathsf{T}}}_{\text{zero padding and slice indexing}} \underbrace{(\mathbf{P}\,\mathbf{K}_1 \otimes \mathbf{K}_2\,\mathbf{P}^{\mathsf{T}} + \sigma^2\mathbf{I})^{-1}(\operatorname{vec}(\mathbf{Y}) - f((\mathbf{X} \times \mathbf{t})) - \boldsymbol{\epsilon})}_{\text{solve inverse matrix-vector product using iterative methods}}.$$

In combination with iterative methods, latent Kronecker structure can be exploited to compute the inverse matrix-vector product using fast MVMs. Further, computations can be cached or amortized to accelerate Bayesian optimization or marginal likelihood optimization [Lin et al., 2023, 2024a,b]. Drawing a single posterior sample takes $\mathcal{O}(n^3 + m^3 + n_*^3)$ time and $\mathcal{O}(n^2 + m^2 + n_*^2)$ space.

## 3   Experiments

We conducted two empirical experiments. In our first experiment, we illustrate the superior scalability of our Latent Kronecker GP (LKGP) compared to naïve Cholesky factorization of the joint covariance matrix. In our second experiment, we predict the final values of partially observed learning curves and evaluate the quality of predictions via mean-square-error (MSE) and log-likelihood (LLH).

**Empirical Time and Space Consumption**   We compared the time and memory requirements for training and prediction of LKGP, which uses iterative methods to exploit latent Kronecker structure, to naïve Cholesky factorization of the joint covariance matrix. To this end, we generated random training data with $n = m \in \{16, 32, ..., 512\}$ and $d = 10$ (see Appendix C for details).

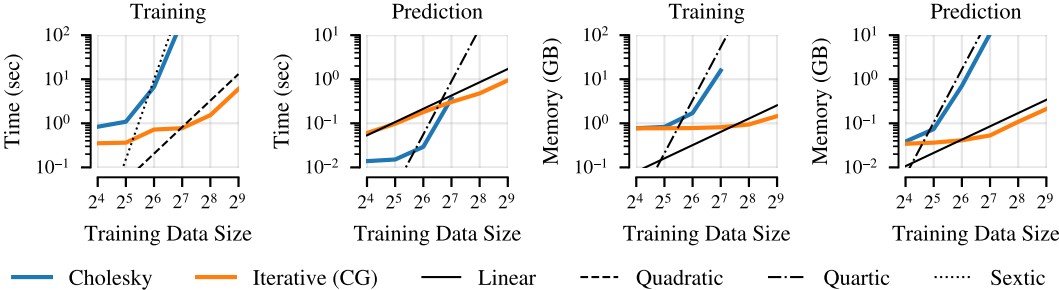

Figure 3: Time and memory consumption as a function of training data size, where size refers to $n = m$. Training consists of optimizing noise $\sigma^2$ and kernel parameters $\boldsymbol{\theta}$. Prediction consists of sampling full learning curves for 512 hyper-parameter configurations. Measurements include constant overheads, such as computations performed by the optimizer or memory reserved by CUDA drivers.

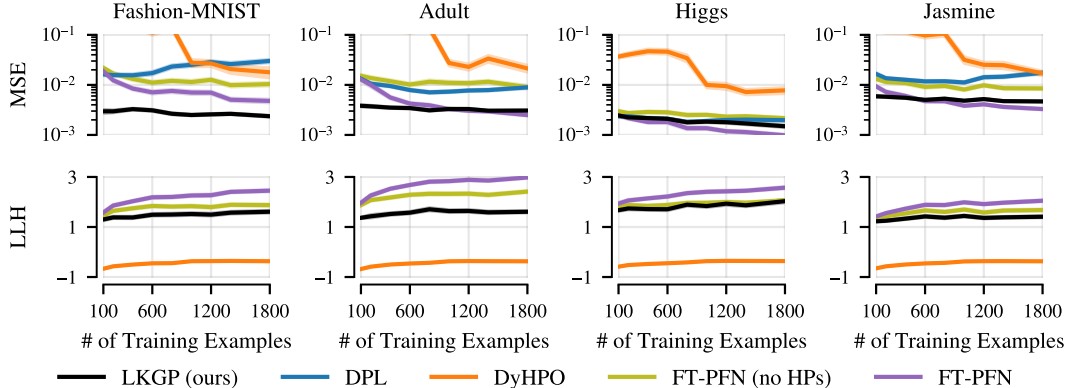

Figure 4: Mean-square-errors (MSE) and log-likelihoods (LLH) of predicted final validation accuracy given partially observed learning curves (mean $\pm$ standard error over 100 random seeds), where # of training examples refers to the total number of observed values across hyper-parameters and progression. Log-likelihood values for DPL are omitted because they are not competitive enough.

Figure 3 visualizes the time and memory consumption as a function of training data size. In general, empirical trends roughly match asymptotic complexities (see Section 2). In particular, LKGP is easily scalable to $n = m = 512$, taking about 6 seconds for training while using less than 2 GB of memory, whereas naïve Cholesky factorization already takes more than 3.5 minutes for training at $n = m = 128$ and runs out of memory at $n = m = 256$. Empirically, in terms of time, LKGP seems to scale *better* than its asymptotic complexity, which is likely due to the iterative solver converging in fewer iterations than would be mathematically required for an exact solution.

**Learning Curve Prediction Quality** We replicated the experiment from Rakotoarison et al. [2024], Section 5.1, who used data from LCBench [Zimmer et al., 2021] to define a learning curve prediction task. In particular, the final validation accuracy is predicted given partially observed learning curves. We used the experimental setup and baseline results from Rakotoarison et al. [2024], which are publicly available on GitHub.[2] The baselines include DPL [Kadra et al., 2023], a neural network ensemble which makes predictions based on power laws; DyHPO [Wistuba et al., 2022], a Gaussian process model equipped with a deep kernel; FT-PFN [Rakotoarison et al., 2024], a Transformer model which is pre-trained on synthetic learning curves and makes predictions via in-context learning; and FT-PFN (no HPs), a version of FT-PFN which does not consider correlations across hyper-parameters.

Figure 4 illustrates that LKGP achieves better or similar MSE compared to other methods and slightly worse LLH compared to FT-PFN. We consider this an impressive result, because LKGP only has access to the partially observed learning curves which are passed to FT-PFN as context at prediction time, uses basic stationary kernels without any inductive bias for modeling learning curves, and only has 10 parameters; whereas the most competitive model, FT-PFN, is pre-trained on millions of samples drawn from a specifically designed learning curve prior, uses a flexible non-Gaussian posterior predictive distribution, and has 14.69 million parameters [Rakotoarison et al., 2024].

## 4 Conclusion

Motivated by AutoML problems, we proposed a joint Gaussian process model over the product space of hyper-parameters and learning curves, which leverages latent Kronecker structure to accelerate computations and reduce memory requirements in the presence of partial learning curve observations. In contrast to existing Gaussian process models with Kronecker structure, our approach deals with missing entries by combining projections and iterative methods. Empirically, we demonstrated that our method has superior scalability than the naïve approach, and that it can match a specialized Transformer model in terms of prediction quality. Future work could investigate specialized kernels and heteroskedastic noise models, which would still be efficient due to the use of iterative methods.

---

[2]`https://github.com/automl/ifBO/tree/icml-2024/src/section5.1`

## Acknowledgments

We thank Herilalaina Rakotoarison for providing details, source code, and data of their learning curve prediction experiment from Rakotoarison et al. [2024], Section 5.1, which allowed us to reproduce it.

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

# A  Related Work

Various methods have been proposed to model learning curves. Swersky et al. [2014] used Gaussian processes with a custom exponential decay kernel, and assumed conditional independence of learning curves given hyper-parameters to improve scalability. Domhan et al. [2015] use a linear combination of assorted parametric basis functions whose parameters are obtained via MCMC. Klein et al. [2017] also use parametric basis functions, but predict their parameters with Bayesian neural networks to introduce correlations across different learning curves. Klein et al. [2020] consider a joint Gaussian process over hyper-parameters and learning curves, but only use a subset of observed values as training data to improve scalability. Wistuba et al. [2022] combine Gaussian processes with a neural network embedding of learning curves. Kadra et al. [2023] use a power law whose coefficients are predicted by an ensemble of neural networks. Adriaensen et al. [2023] train a Transformer on synthetic learning curves sampled from a prior defined by parametric functions, and make predictions via in-context learning. Rakotoarison et al. [2024] use the same approach but also integrate hyper-parameters into the tokens to introduce correlations across learning curves.

# B  Implementation Details

We implemented LKGP using GPyTorch [Gardner et al., 2018] and performed all experiments in double floating point precision. Further details are discussed below.

**Model Specification and Prior Distributions**   The mean of the joint model is assumed to be zero. We used a RBF kernel over hyper-parameters $\mathbf{x} \in \mathbb{R}^d$ with a length scale parameter per dimension. Following Hvarfner et al. [2024], we set the prior over length scales to $\log \mathcal{N}(\sqrt{2} + 0.5 * \log d, \sqrt{3})$. For the learning curve progression $t \in \mathbb{R}$, we used a Matérn-$5/2$ kernel with a scalar length scale and a scalar output scale, both without any prior. We used a homoskedastic Gaussian likelihood with a scalar noise variance parameter whose prior is $\log \mathcal{N}(-4, 1)$. As a result, there are 10 model parameters which we optimized by maximizing the marginal likelihood plus priors using L-BFGS.

**Input and Output Transformations**   We normalize every $\mathbf{x} \in \mathbb{R}^d$ to the unit hypercube. To this end, for each dimension, we subtract the minimum and divide by the difference between maximum and minimum of that dimension in the training data. We transform every $\mathbf{t} = \{t_1, \ldots, t_m\}$ by first applying the logarithm to $t_1, \ldots, t_m$ before subtracting $\log t_1$ and dividing by $\log t_m - \log t_1$, such that the result is the unit interval with logarithmic spacing. The outputs $\mathbf{Y}$ are standardized by first subtracting the largest value in $\mathbf{Y}$ and then dividing by the standard deviation over all elements in $\mathbf{Y}$.

**Iterative Methods for Posterior Inference**   We used batched conjugate gradients [Gardner et al., 2018] with a relative residual norm tolerance of 0.01, and a maximum number of 10000 iterations. If the kernel over the progression $t$ is stationary and $t$ is logged and modeled on a regular grid then the resulting Toeplitz structure can be used to further speed up MVMs, similar to SKI [Wilson and Nickisch, 2015]. However, our $\log t$ transformation interferes with the original regularly spaced grid. Furthermore, future modeling improvements might include specialized, non-stationary kernels that incorporate certain decay assumptions like the ones explored by Swersky et al. [2014] and Klein et al. [2020], which would also prevent the use of Toeplitz structure.

# C  Experiment Details

**Asymptotic Time and Memory Consumption**   We generated random training data for different $n = m \in \{16, 32, 64, 128, 256, 512\}$ and $d = 10$. Each entry in $\mathbf{X} \in \mathbb{R}^{n \times d}$ was drawn uniformly at random from the unit interval, and each entry in $\mathbf{Y} \in \mathbb{R}^{n \times m}$ was drawn from a standard normal distribution. All entries in $\mathbf{X}$ and $\mathbf{Y}$ were sampled independently. The values in $\mathbf{t} \in \mathbb{R}^m$ were set to represent the unit interval with linear spacing. No missing data was introduced. Kernel evaluations take $\mathcal{O}(d)$ time. All computations and resource measurements were performed on a V100 GPU.

**Learning Curve Prediction Quality**   We adopted the experimental setup and baseline results from Rakotoarison et al. [2024], Section 5.1, but instead of reporting the median over all tasks, we illustrate the performance per task in Figure 4. Metrics are computed over 100 different random seeds, which have also been published by Rakotoarison et al. [2024], such that we were able to match them.

