# OpenReview forum: "Scaling Gaussian Processes for Learning Curve Prediction via Latent Kronecker Structure"
_NeurIPS.cc/2024/Workshop/BDU — NeurIPS BDU Workshop 2024 Poster_

### Official Review · Reviewer_ELhK · 2024-09-26

**Rating:** 8
**Confidence:** 3

**Review:**

Summary:
----------
The authors propose to leverage a projection of a latent Kronecker structure to preserve the advantages brought by using kernels, but also handle missing values in inference of joint models of learning curves and hyperparameters for AutoML tasks. This roughly halves the order of time and storage space needed.

Main Review:
----------
Quality and clarity: The paper is well-written, exposition and results are well-explained.

Figure 1 shows good results.
Figure 3 is convincing, as it shows a clear advantage of LKGP over the naive approach.
I would be nice to have at least one line (in the appendix) mentioning how the choice of d impacts the results, if at all.

It would be interesting how the kernel choice affects the results and what motivated the choices of RBF and Matérn kernel.
The enable reproducibility, the authors code for LKGP should be made available online.

I would like to know the motivation for illustrating the performance per task instead of reporting the median over all tasks in Figure 4.

Typos
--------
In l. 211 the citation is formatted differently that e.g. in l. 196.

Citations
--------
The "a." in l. 127 should be removed.
The reference in ll. 142-143 is missing a venue.
In ll. 160 and 162 it should be "Gaussian" and "Bayesian".
The capitalization of titles is inconsistent: for most titles, almost all words are capitalized, but not for the reference in ll. 159-161.
The line break in ll. 178-179 is unfortunate, one should probably define a custom hyphenation for that word.

---

### Official Review · Reviewer_JrG9 · 2024-09-26

**Rating:** 6
**Confidence:** 3

**Review:**

The overall quality of the paper is good, utilizing latent Kronecker structure to over come the complexity issues in traditional Gaussian processes. The paper is generally clearly written, but would benefit from more detailed mathematical presentations. The idea of latent Kronecker structure is interesting, although not so ground-breaking. The paper shows its strengths in learning curve prediction tasks, but its scalability and applications to other practical tasks are unknown.

Pros:
* Propose latent Kronecker structure, which is simple but effective.

Cons:
* Limited exploration of specialized kernels or heteroskedastic noise models.
* More diverse datasets could strengthen the empirical validation.
* Lacks discussions on scalability and applications to other practical tasks in addition to cure prediction.

---

### Decision · Program_Chairs · 2024-10-09

Accept (Poster)